# A Clinical Validation of a Diagnostic Test for Esophageal Adenocarcinoma Based on a Novel Serum Glycoprotein Biomarker Panel: PromarkerEso

**DOI:** 10.3390/proteomes13020023

**Published:** 2025-06-04

**Authors:** Jordana Sheahan, Iris Wang, Peter Galettis, David I. Watson, Virendra Joshi, Michelle M. Hill, Richard Lipscombe, Kirsten Peters, Scott Bringans

**Affiliations:** 1Proteomics International, Broadway, P.O. Box 3008, Perth 6009, WA, Australia; jordana@proteomics.com.au (J.S.); iris@proteomics.com.au (I.W.); peter@proteomics.com.au (P.G.); richard@proteomics.com.au (R.L.); kirsten@proteomics.com.au (K.P.); 2Department of Surgery, Flinders Medical Centre, College of Medicine and Public Health, Flinders University, Adelaide 5042, SA, Australia; david.watson@flinders.edu.au; 3School of Medicine, Emory University, Atlanta, GA 30322, USA; virendra.joshi@emory.edu; 4QIMR Berghofer Medical Research Institute, Brisbane 4006, QLD, Australia; michelle@proseekbio.com

**Keywords:** MRM-MS, glycoprotein, biomarkers, LeMBA, esophageal adenocarcinoma (EAC), Gastrointestinal diseases, targeted mass spectrometry, cancer diagnosis, PromarkerEso

## Abstract

Background: Esophageal adenocarcinoma (EAC) diagnosis involves invasive and expensive endoscopy with biopsy, but rising EAC incidence has not been reduced by increased surveillance. This study aimed to develop and clinically validate a novel glycoprotein biomarker blood test for EAC, named PromarkerEso. Methods: Serum glycoprotein relative concentrations were measured using a lectin-based magnetic bead array pulldown method, with multiple reaction monitoring mass spectrometry in 259 samples across three independent cohorts. A panel of glycoproteins: alpha-1-antitrypsin, alpha-1-antichymotrypsin, complement C9 and plasma kallikrein, were combined with clinical factors (age, sex and BMI) in an algorithm to categorize the samples by the risk of EAC. Results: PromarkerEso demonstrated a strong discrimination of EAC from the controls (area under the curve (AUC) of 0.91 in the development cohort and 0.82 and 0.98 in the validation cohorts). The test exhibited a high sensitivity for EAC (98% in the development cohort, and 99.9% and 91% in the validation cohorts) and a high specificity (88% in the development cohort, and 86% and 99% in the validation cohorts). PromarkerEso identified individuals with and without EAC (96% and 95% positive and negative predictive values). Conclusions: This less invasive approach for EAC detection with the novel combination of these glycoprotein biomarkers and clinical factors coalesces in a potential step toward improved diagnosis.

## 1. Introduction

Esophageal cancer is the seventh leading cause of cancer-related deaths and the eleventh most prevalent cancer worldwide, with approximately 445,000 deaths in 2022 [1]. Esophageal cancer consists of two main subtypes: esophageal squamous cell carcinoma and esophageal adenocarcinoma (EAC). The incidence of EAC in Western countries has increased by 600% since the 1970s, now accounting for up to 85% of cases, while squamous cell carcinoma is more common in East Asia and the Middle East. The prognosis of EAC is poor, with a five-year survival rate of less than 20% and a median survival time of less than one year [2]. EAC develops as a neoplastic change in epithelium in the lower esophagus near the gastroesophageal junction [3], with the current gold standard for a diagnosis being an endoscopy followed by a biopsy. A major challenge in managing EAC is that it is asymptomatic in its early stages, leading to a delayed diagnosis. As a result, many patients are diagnosed at advanced stages, limiting early intervention opportunities and significantly impacting prognosis [4].

The known risk factors for EAC are gastroesophageal reflux disease (GERD) and Barrett’s esophagus (BE), a pre-malignant condition known to evolve through dysplasia into invasive carcinoma [5]. The current clinical guidelines recommend regular endoscopic surveillance of individuals with BE [6,7]; however, frequently BE does not evolve into EAC, with the annual progression rates as low as 0.1% to 0.3% [8]. Similarly, only 66% of EAC patients report having symptoms of GERD, whilst over 90% of EAC cases lack a prior history of BE [9,10]. Notably, endoscopic procedures are invasive and require hospital facilities and the risk of misdiagnoses may lead to either overtreatment or undertreatment of many patients [11,12]. Existing surveillance strategies are failing to curb the rising incidence of EAC [13], emphasizing the need for improved diagnostic tools with more efficient strategies to address the growing burden of this disease.

There are several diagnostic biomarker tests available for EAC, but these are invasive, requiring tissue specimens and the access to cancerous cells. Currently, there are no blood-based biomarker tests clinically available for the diagnosis of EAC, however, such biomarkers offer potential as a non-invasive alternative with several glycoproteins showing promise [14,15,16], along with several markers related to genomic instability, metabolite abnormalities and miRNA [17,18,19]. Nonetheless, these non-invasive biomarker tests are yet to reach clinical viability and there is still no alternative to endoscopic procedures for the detection of cancer.

The aim of this present study was to develop and validate a non-invasive test for the detection of EAC. To achieve this, a panel of serum glycoproteoform biomarkers, previously identified by Shah et al. [20], were measured using a lectin-based magnetic bead array (LeMBA) pulldown method coupled with targeted mass spectrometry [21]. The advantage of the LeMBA methodology is the selectivity and enrichment it offers for the specific glycoproteins associated with EAC and the effective removal of non-specific proteins in serum, improving mass spectrometry detection and sensitivity. The previous methods utilized several different lectins independently that pulled down distinct sets of glycoproteins. Jacalin was selected for this assay development as representative of lectin pulldown of glycoproteins that correlate with esophageal adenocarcinoma as shown by Shah et al. [20]. This current study examined two independent cohorts comprising samples with EAC and negative controls (NC) confirmed not to have had EAC previously investigated by Shah et al. In addition, a third independent cohort with general population (GenPop) controls and samples with EAC was utilized for further validation. The outcome is the development of a novel validated diagnostic blood test, PromarkerEso, which measures four specific glycoproteins coupled with simple clinical factors (age, sex and BMI) that when combined detect the presence of EAC.

## 2. Materials and Methods

### 2.1. Experimental Design

This study analysed the serum samples from three independent cohorts: the Progression of Barrett’s Oesophagus to Cancer Network (PROBE-NET, Sydney NSW, Brisbane QLD, Adelaide SA and Melbourne VIC, Australia) study, the Ochsner Health System (Ochsner, New Orleans, LO, USA) and the Victorian Cancer Biobank (VCB, Melbourne, VIC, Australia) (Figure 1). The details of the PROBE-NET and Ochsner cohorts have been published elsewhere [20,22]. The development cohort comprised 103 samples from the PROBE-NET study, collected between 2008 and 2016, including 60 participants diagnosed with EAC based on the endoscopy and histopathology and 43 controls confirmed as not having EAC (or BE) at the endoscopy (NC). Of these, 15 samples were excluded due to missing BMI data (*n* = 12 EAC and *n* = 3 NC). The first validation cohort contained 24 participants from the Ochsner study, collected 2014–2015, with 14 confirmed via endoscopy to be NC and 10 confirmed by endoscopy as EAC. The second validation cohort comprised 166 samples from the VCB, which recruited participants from five hospital-integrated biobanks through a central lead agency in Melbourne, Australia. A total of 66 of these samples were confirmed to have had EAC via endoscopy and 100 GenPop healthy donor controls with no known history of cancer, all of which were recruited through the biobank consortium and fulfilled the following inclusion criteria: were above the age of 18 years, not pregnant or lactating and able to give informed consent. Some samples were excluded due to missing BMI data (*n* = 8 EAC and *n* = 10 GenPop controls) and one further GenPop control excluded due to incomplete analytical data. Whilst a pathological diagnosis was available for all the samples (namely the presence or absence of EAC), information on the stage of EAC was not consistently available. A similar procedure of the serum collection was followed across all the study sites with the serum collected according to standard protocols [20,22]. All the serum samples were stored at −80 °C prior to study. Additionally, de-identified clinical information including age, body mass index (BMI) and sex were available for the three cohorts. This data was integrated with glycoprotein relative concentration measurements for analysis and model development. Informed consent was obtained from all the participants at the point of recruitment for each separate study and ethics was obtained by each participating institution, with the validation of the VCB cohort by Proteomics International approved by the Bellberry ethics committee (Bellberry Limited NHMRC-registered EC00372 HREC 2023-01-006-A-4).

### 2.2. Targeted Mass Spectrometry Analysis

The relative concentrations of nine candidate serum glycoproteins were assessed via a lectin-based magnetic bead array (LeMBA) pulldown method and quantified using multiple reaction monitoring mass spectrometry (MRM-MS) based on the methods previously described [16,20,21]. Optimizations for this study included reduced digestion time, use of low microliter flow rates, more efficient sample processing and minimal sample transfer steps with the optimized method described below.

The lectin Jacalin (jacalin from Artocarpus integrifolia) was conjugated to MyOne Tosyl-activated Dynabeads^®^ (Thermo Fisher Scientific, Waltham, MA, USA) as previously described [23] at 50 µg lectin per mg of bead. Serum (6 µL) was added to 114 µL of denaturing and reduction solution (25 mM Tris buffer pH 7.4, 1% *w*/*v* sodium dodecyl sulphate (SDS), 3% *v*/*v* Triton X-100, 25 mM dithiothreitol) at 37 °C for 30 min, then alkylated using 5 µL iodoacetic acid (100 mM) at room temperature in the dark for 30 min. The denatured serum (100 µL out of 125 µL reaction volume) was then incubated with the Jacalin beads (50 µL beads at 100 mg/mL in 20 mM Tris buffer pH 7.4, 300 mM NaCl, 1 mM CaCl_2_, 1 mM MnCl_2_, 1% Triton X-100) at 4 °C for 1 h on a plate shaker at 500 rpm. After incubation, the beads were washed (six times with 50 mM ammonium bicarbonate on a plate washer) and resuspended in 100 µL 50 mM ammonium bicarbonate. The beads then underwent a trypsin digestion (10 µL of 0.05 µg/µL Rapid trypsin gold (Promega, Madison, WI, USA) in 50 mM acetic acid for 1 h at 70 °C. Digestion was quenched with 6 µL of 2% formic acid. This liquid was removed from the beads for mass spectrometry analysis as described below.

In total, 25 peptides were measured across the nine candidate glycoproteins (detailed in Appendix A). In brief, Ovalbumin protein (15 µg) was spiked into each sample as an internal calibration standard before LeMBA pulldown and used to normalize the samples. The targeted transitions were scheduled for each peptide with a retention time window of 40 s. Each assay measured summed peak areas of the target peptide transitions of individual plasma samples against the summed peak area of the three transitions of the Ovalbumin peptide (GGLEPINFQTAADQAR) to calculate the peak area ratios for each of the biomarkers. The peptide data presented therefore reflect the relative concentration of a glycoprotein biomarker between samples.

The samples were injected from the digested bead solution (20 µL) and analysed using a 6500+ triple quadrupole mass spectrometer (Sciex, Framingham, MA, USA) coupled to an Ultimate 3000 HPLC (Thermofisher Scientific) with a Pepmap100 Acclaim C18 column (0.3 × 150 mm, Thermofisher Scientific) at a flow rate of 10 µL/min and 2% (*v*/*v*) acetonitrile with 0.1% (*v*/*v*) formic acid (mobile phase A) and 98% (*v*/*v*) acetonitrile with 0.1% (*v*/*v*) formic acid (mobile phase B). The LCMS parameters are detailed in Appendix A.

For inter-batch quality control, a reference serum aliquot was analysed in four replicates alongside every batch of cohort samples. The coefficient of variation (%CV) was calculated across the four intra-assay replicates for each analysed peptide with greater than two-thirds of the peptide assays required to have a CV < 30% for that batch to be acceptable for data analysis. Reproducibility of the assay was determined by a separate assessment of the inter-assay CV% for each of the peptides of the assay for 48 replicate preparations of the fixed reference serum across 6 assay runs. (Data shown in Appendix A).

### 2.3. Statistical Rationale

Peptide relative concentrations were calculated as the ratio of each peptide signal area over the calibrant (ovalbumin) peptide signal area. Natural logarithmic transformation was applied to non-normally distributed variables. Clinical characteristics and peptide ratios between EAC and the controls within the cohorts were compared using the Mann–Whitney U test (Appendix A). Kendall’s Tau was used to assess intra- and inter- protein multicollinearity; this ensured only one peptide was modelled per protein and confirmed the assumption of covariate independence was met (Appendix A).

A parsimonious model for predicting binary outcomes (negative for EAC controls vs. EAC positive) was developed using stepwise logistic regression, both forward and backward, to select covariates from 25 peptides and clinical variables (age, sex and BMI) for adjustment. A variable was only included if it added significant independent value to the model, as determined by likelihood ratio *p* values. Other models were considered such as machine learning model approaches; however these were deemed less suitable in terms of interpretability and performance in this context.

The model was developed and trained on the PROBE-NET cohort and validated with the Ochsner and VCB cohorts. Model outputs were predicted probabilities for having EAC ranging from 0% to 100% and categorized as low-, moderate- or high-risk as defined by cutoffs to maximize the test performance. The cutoffs were set at 20% and 75% to separate people into low- (<20%), moderate- (between 20% and 75%) and high-risk (>75%) for EAC which maximized the sensitivity at the lower cutoff and maximized the specificity at the upper cutoff.

The performance of the model was assessed using the area under the receiver operator curve (AUC) and an optimism-corrected AUC for internal validity, which was determined using the bootstrapping of 1000 samples. The predicted outcomes were assessed for association with actual outcomes using Fisher’s exact test. A calibration of the model was assessed graphically by plotting the predicted probabilities against the actual outcomes (Appendix A). All the statistical analyses were performed in R (version 4.4.1) using RStudio software (version RStudio 2024.04.2 + 764). A two tailed level of significance of *p* < 0.05 was used throughout.

## 3. Results

### 3.1. Targeted Mass Spectrometry

During the initial testing, the serum glycoproteins were measured with an MRM analysis of the target peptides representing proteins previously reported from the discovery experiments [16]. The MRM analysis measured these 25 peptides from nine proteins from the Jacalin lectin bead pulldown. The transitions are described in Appendix A. Reproducibility of the assay in development was determined by inter-assay assessment across 48 replicates from six independent assays, with mean, standard deviation and %CV from the peak area ratio to the ovalbumin internal standard shown in Appendix A for each peptide. Preliminary bivariate comparison identified six out of nine candidate glycoproteins had a significant (*p* value < 0.05) log2 fold change in the development cohort PROBE-NET in the EAC samples compared to the NC samples (Figure 2).

### 3.2. Participant Characteristics

The key clinical characteristics including the age, body mass index (BMI) and sex were compared for distribution differences between EAC and the controls (Table 1). The participants with EAC were significantly older and more often male compared to the NC and GenPop controls in PROBE-NET and VCB, respectively (all *p* < 0.001). No significant differences in age or sex were seen in the Ochsner cohort. The BMI was significantly higher in EAC compared to controls in the VCB cohort (*p* = 0.016); however no BMI differences were observed in the PROBE-NET or Ochsner cohorts.

### 3.3. Development of the Assay Algorithm

The algorithm was developed as a multiple logistic regression with optimization for maximum sensitivity and specificity. The model covariates were selected from the 25 peptides, which were representative of the nine glycoproteins and clinical covariates using the stepwise logistic regression approach. While the six glycoproteins with bivariate significance between EAC and NC were considered as important covariates for this model (Figure 2), all the nine glycoproteins were evaluated for covariate selection due to potential inter-glycoprotein relationships resulting in an optimized model. Hence, the best fit included four peptides representing the glycoproteins: alpha-1-antitrypsin, complement C9, alpha-1-antichymotrypsin, and plasma kallikrein, together with age, sex and BMI. A bivariate analysis of these peptides demonstrated significant differences between EAC and the controls (*p* < 0.005) for alpha-1-antitrypsin, complement C9 and plasma kallikrein in PROBE-NET (Table 1). An assessment for the multicollinearity of these peptides in PROBE-NET demonstrated a significant but moderate correlation between alpha-1-antitrypsin, alpha-1-antichymotrypsin and complement C9; however there was little correlation with plasma kallikrein (Appendix A). Biomarkers provided significantly improved discrimination of controls from EAC compared to clinical factors of age, sex and BMI alone (delta-AUC 0.07, *p* < 0.0003).

### 3.4. Performance of the PromarkerEso Assay

The performance of the PromarkerEso assay was assessed using the AUC in the development cohort and validated in two independent cohorts (Table 2 and Figure 3). The performance of the model in PROBE-NET was strong with an AUC of 0.91 (95% CI 0.84, 0.96) (Figure 3). The discrimination accuracy of the assay in the development cohort was also high, with 98% sensitivity/96% NPV at the lower cutoff and 88% specificity/87% PPV at the upper cutoff (Table 2).

A clinical validation was performed in two independent cohorts. In the Ochsner cohort, the algorithm performed well with a strong overall AUC 0.82 (95% CI = 0.64, 0.98) despite the small sample size. In the VCB cohort, the AUC demonstrated excellent discrimination of 0.98 (95% CI 0.96, 1.00). The model discrimination was well validated with 91% sensitivity/95% NPV at the lower cutoff and 99% specificity/98% PPV at the upper cutoff.

A comparison of the test predictions with the actual diagnosis of EAC demonstrated there was significant discrimination between EAC and the controls across all the cohorts (*p* < 0.013) (Table 3 and Figure 4A–C). The Low-Risk category reliably discriminated those individuals without EAC, as evidenced by a few true positives in the Low-Risk category in the PROBE-NET and VCB cohorts (1 true positive out of 48 total and 5 true positives out of 58 total respectively) and a complete absence of true positives in the Low-Risk category in the Ochsner cohort. The High-Risk category complements this by effectively capturing the individuals with EAC as demonstrated with 33 true positives out of 48 total in PROBE-NET falling into this category. This trend is extended in the validation cohorts with 6 true positives out of 10 (Ochsner) and 43 out of 58 (VCB) true positives, respectively.

The model demonstrated reasonable calibration between the actual observations and the predicted probabilities across the cohorts with a minor deviation from the ideal indicating a slight over-estimation in PROBE-NET and Ochsner and an under-estimation in VCB (Appendix A).

## 4. Discussion

The PromarkerEso test offers a potential advancement in the detection of EAC via a novel combination of glycoprotein biomarkers. The MRM assay measures four proteotypic peptides that define the specific glycoproteoforms of the biomarkers pulled down with the lectin Jacalin. Jacalin is known to bind to specific O-linked galactose structures (Galα1-6GalNAc and Galβ1-3GalNAc) [24]. These O-linked galactose structures have been previously associated with esophageal cancers and are of interest due to both changes in the glycosylation patterns and the proteins themselves [12,16,25]. Altered protein glycosylation patterns have been associated with EAC and signaling pathways involved in malignant cell transformation [15]. These aberrant glycosylation modifications can affect the protein function, including those involved in cell adhesion, migration, and proliferation, ultimately contributing to the initiation and progression of cancer. Dysregulated glycosylation processes may therefore play a role in the molecular mechanisms driving EAC, influencing tumor development and metastasis through the modulation of various signaling networks.

The biomarker proteins identified in this study and represented by the glycosylated forms measured in PromarkerEso have known links to EAC. Analysis of biopsy samples from EAC and BE participants in the Ochsner cohort by Shah et al. revealed the presence of complement C9 [20], which is a subunit of the membrane attack complex forming pores in cell plasma membrane of cells targeted by immune response [26]. This finding aligns with previous research indicating complement pathway dysregulation in EAC pathogenesis [27]. Serum and gastric secretion concentrations of serine protease inhibitors alpha-1-antitrypsin and alpha-1-antichymotrypsin have been found to significantly increase during the early stages of similar cancers such as gastric and colorectal cancer [28]. Alpha-1-antitrypsin, known for its inhibitory effect on neutrophil elastase [29], is also elevated in esophageal squamous cell carcinoma, the squamous cell variant of EAC. Elevated plasma levels of alpha-1-antichymotrypsin are associated with similar disease processes in esophageal squamous cell carcinoma [30]. Plasma kallikrein, a component of the human kallikrein-related peptidases has been found to have aberrant abundance in multiple cancer types, including GI cancers, which affects the cascade of tumor management, such as trypsin and chymotrypsin-like serine peptidases [31].

The developed PromarkerEso test has demonstrated strong efficacy across three well curated cohorts in distinguishing between general population controls, negative controls and individuals diagnosed with EAC. Notably, the algorithm showed minimal evidence of overfitting, underscoring its robustness and generalizability across diverse populations. Following strong performance in the development cohort, the PromarkerEso test achieved excellent results in the validation cohorts, where the sensitivity and specificity reached 99.9% and 85.7% in Ochsner and 91.4% and 98.9% in VCB, respectively. These results are comparable to other novel clinical tests in development [32,33,34].

The three-stage risk category design of PromarkerEso enhances predictive accuracy while mitigating uncertainty in mid-range probability predictions, with elevated sensitivity and specificity at lower and upper cutoffs respectively. The Low-Risk category reliably discriminates those individuals without EAC (true negatives). In contrast, the High-Risk category effectively captures individuals with EAC across all cohorts. Similarly, the Moderate-Risk category is predominantly composed of true positives; however, as illustrated in the classification table (Table 3), there is a small proportion of true negatives (28% in PROBE-NET, 36% in Ochsner and 1.1% in VCB).

The limitations to this study were considered when interpreting results. The demographic distributions of the clinical cohorts differed significantly, with age, sex and BMI adjustments made during the model development to reduce the potential biases. The unequal sex distribution across cohorts—where the male sex is overrepresented in EAC—is acknowledged, which may limit the generalizability of PromarkerEso in female populations. Further sex and age-matched studies are required to rule out confounding due to this imbalance in addition to adjustment for sex bias by the inclusion of sex in the PromarkerEso model as a covariate. Additionally, the general population controls in the VCB cohort were not definitively confirmed negative for EAC through endoscopy; hence the presence of EAC cannot be completely ruled out, although it seems highly unlikely. Furthermore, the outcomes from the general population controls serve as a useful test of the generalizability of PromarkerEso to healthy individuals, without the need to assume the presence of clinical risk factors typically warranting endoscopy, such as GERD. The lectin Jacalin and lectins in general are not universal glyco-group binders, rather Jacalin binds specific O-linked galactose structures. While it is understood that this was by no means an exhaustive analysis for potential glycoprotein markers, Jacalin was selected for this assay development as it had the most suitable discrimination for esophageal adenocarcinoma and best analytical reproducibility. Significant differences in glycoprotein relative concentrations (measured as median peptide ratios) were observed across the cohorts, and this needs to be explored; however, it did not indicate model overfitting, as there was no evidence of performance loss during validation. As ethnicity information was not collected from/provided by the participants who retrospectively contributed samples, it was not possible to adjust for the ethnic differences in the cohort. The performance of the biomarkers identified in assessing the cancer stage was beyond the scope of this study; however, this analysis will be important to better understand the clinical utility of the test in detecting EAC at an early stage.

PromarkerEso offers the potential for integration into the current standard of care, with blood testing conducted prior to or while awaiting referral for routine endoscopy and biopsy. The interpretation of test results could then enable individuals with a Low-Risk outcome to avoid unnecessary endoscopies, while those classified as Moderate Risk could proceed with recommended routine endoscopy and biopsy. High-Risk individuals would be prioritized for endoscopic testing, effectively reducing wait times and overall medical care burden.

PromarkerEso demonstrates a consistent and reliable performance in detecting esophageal adenocarcinoma across diverse populations. The novelty of this study is highlighted by the unique combination of these four glycoprotein biomarkers and three clinical factors that together as a blood test offers an advantage over standard care as it is more convenient and less invasive for patients. Whilst PromarkerEso has been validated in general healthy populations, future studies are required to assess the performance of PromarkerEso in different stages of cancer and applicability in broader demographic groups. It will also be interesting to assess the accuracy of the biomarkers as a potential surveillance option after treatment for EAC. Nonetheless, the results of this study highlight the potential of PromarkerEso as a tool to enhance the standard diagnostic framework for EAC, and a step toward improving patient care in this challenging disease.

This article is a revised and expanded version of a poster entitled “Validation of PromarkerEso, a Diagnostic Blood Test to Identify Esophageal Adenocarcinoma”, which was presented at the 20th ISDE World Congress for Esophageal Disease in Edinburgh, Scotland in 2024 [35].

## Figures and Tables

**Figure 1 proteomes-13-00023-f001:**
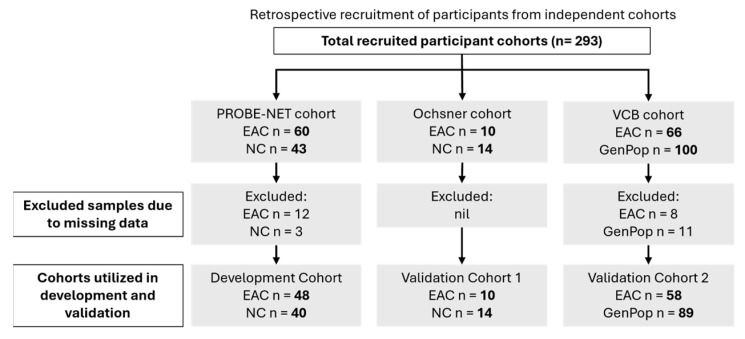
Participant recruitment details and allocations of samples for modelling. EAC = esophageal adenocarcinoma, GenPop = general population controls, NC = control by negative endoscopy.

**Figure 2 proteomes-13-00023-f002:**
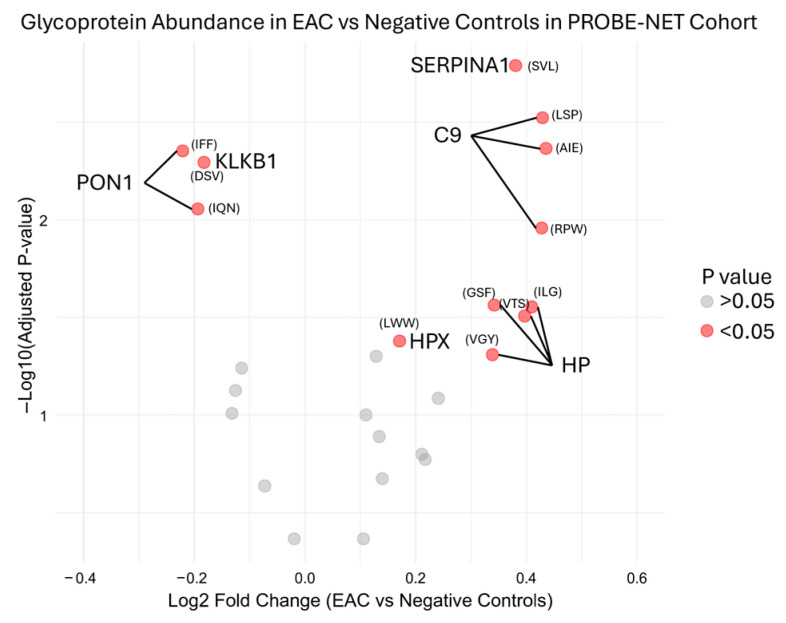
Differential glycoprotein fold change abundance in esophageal adenocarcinoma compared to negative controls in PROBE-NET cohort for development of assay. Relative peptide concentration is represented here with first three letters of peptide sequence by log2 fold change of peak area ratio of each glycoprotein over internal standard. Full sequences are available in Appendix A. *p* values are from bivariate analysis using Mann–Whitney U Testing between EAC and negative controls. EAC = esophageal adenocarcinoma, SERPINA1 = alpha-1-antitrypsin, C9 = complement C9, HP = haptoglobin, HPX = hemopexin, PON1 = serum paraoxonase/arylesterase, KLKB1 = plasma kallikrein.

**Figure 3 proteomes-13-00023-f003:**
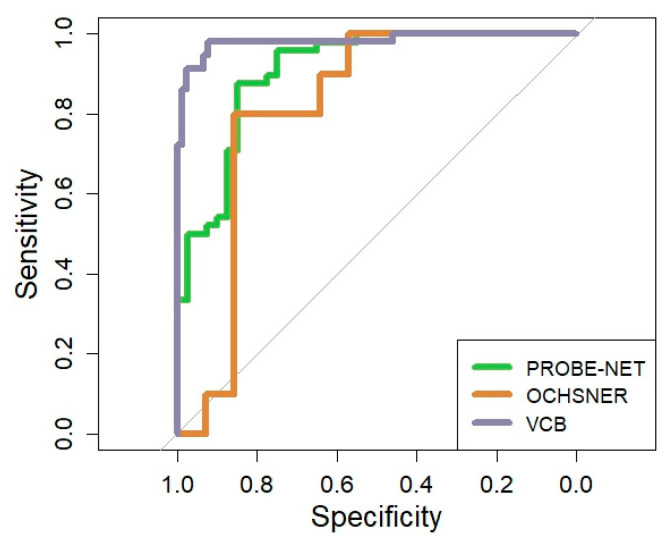
Overall Receiver Operator Curve for PromarkerEso assay in development cohort and validation cohorts. The grey line represents the line of no discrimination (area under the curve = 0.5).

**Figure 4 proteomes-13-00023-f004:**
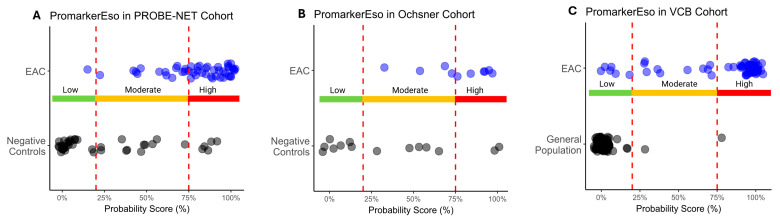
Distribution of esophageal adenocarcinoma (EAC) and control sample probability scores as classified by PromarkerEso into Low-, Moderate- and High-Risk categories in development cohort (**A**) PROBE-NET and validation cohorts (**B**) Ochsner and (**C**) VCB. Actual outcomes are represented as blue dots (EAC) and black dots (Negative Controls or General Population). Lower and upper cutoffs (20% and 75% respectively) are represented by red dotted lines.

**Table 1 proteomes-13-00023-t001:** Comparison of demographics and key biomarkers included in algorithm in PROBE-NET, Ochsner and VCB cohorts as selected by stepwise logistic regression. Distribution values described in median ± median absolute value unless described otherwise. *p* values are from bivariate testing conducted between negative controls (NC) or general population controls (GenPop) vs. EAC across three cohorts using Mann–Whitney U Test.

	PROBE-NET *n* = 88			Ochsner *n* = 24			VCB *n* = 147		
Demographic	NC (*n* = 40)	EAC (*n* = 48)	*p* Value	NC (*n* = 14)	EAC (*n* = 10)	*p* Value	GenPop (*n* = 89)	EAC (*n* = 58)	*p* Value
Age (years)	57.0 ± 13.3	66.0 ± 11.9	0.0003	65.5 ± 16.3	62.5 ± 3.0	0.39	35.0 ± 11.9	65.5 ± 11.1	<0.0001
Sex (% male)	35.0	83.3	<0.0001	79.0	80.0	0.97	19.1	91.1	<0.0001
BMI (kg/m^2^)	27.7 ± 7.1	26.8 ± 2.7	0.91	31.1 ± 6.7	28.5 ± 3.8	0.17	25.1 ± 4.7	27.7 ± 4.5	0.016
Key Biomarkers									
Alpha-1-antitrypsin	1.63 ± 0.47	1.99 ± 0.71	0.0016	1.67 ± 0.60	3.18 ± 0.57	0.0004	3.10 ± 1.20	5.92 ± 2.74	<0.0001
Alpha-1-antichymotrypsin	0.85 ± 0.19	0.97 ± 0.39	0.17	1.06 ± 0.40	1.58 ± 0.36	0.0017	1.85 ± 0.71	2.07 ± 0.76	0.27
Complement C9	0.47 ± 0.12	0.59 ± 0.25	0.003	0.58 ± 0.21	0.88 ± 0.25	0.0026	0.85 ± 0.34	1.63 ± 0.69	<0.0001
Plasma kallikrein	0.04 ± 0.01	0.04 ± 0.01	0.005	0.07 ± 0.02	0.06 ± 0.01	0.68	0.10 ± 0.05	0.08 ± 0.03	0.0045

**Table 2 proteomes-13-00023-t002:** Performance metrics of PromarkerEso assay in development cohort (PROBE-NET) and validation cohorts (Ochsner and VCB). NC = control by negative endoscopy test; GenPop = General population controls; AUC = area under the curve; PPV = positive predictive value; NPV = negative predictive value.

Performance Measure	PROBE-NET	Ochsner	VCB
Number of Subjects	88 (EAC = 48, NC = 40)	24 (EAC = 10, NC = 14)	147 (EAC = 58, GenPop = 89)
Discrimination:			
AUC (95%CI)	0.91 (0.84–0.96)	0.82 (0.64–0.98)	0.98 (0.96–1.00)
Model at 20% cutoff:			
Sensitivity (%)	97.9	99.9	91.4
Specificity (%)	60.0	50.0	97.8
PPV (%)	74.6	58.8	96.4
NPV (%)	96.0	99.9	94.6
Model at 75% cutoff:			
Sensitivity (%)	68.8	60.0	74.1
Specificity (%)	87.5	85.7	98.9
PPV (%)	86.8	75.0	97.7
NPV (%)	70.0	75.0	85.4

**Table 3 proteomes-13-00023-t003:** Comparison of PromarkerEso test classification to actual disease status. EAC = esophageal adenocarcinoma, GenPop = general population controls, NC = control by negative endoscopy test.

PromarkerEso Test Classification						
Cohort	Actual Outcome	Low Risk	Moderate Risk	High Risk	Total	Fisher’s Exact Test (*p* Value)
PROBE-NET	EAC	1	14	33	48	<0.0001
	NC	24	11	5	40	
	Total	25	25	38	88	
Ochsner	EAC	0	4	6	10	0.013
	NC	7	5	2	14	
	Total	7	9	8	24	
VCB	EAC	5	10	43	58	<0.0001
	GenPop	87	1	1	89	
	Total	92	11	44	147	

## Data Availability

All processed data is contained within this manuscript and Appendix A. All the raw data has been deposited to the ProteomeXchange Consortium [36] via PASSEL [37] with the dataset identifier PASS05904 (Server name: ftp.peptideatlas.org; Full URL: http://www.peptideatlas.org/PASS/PASS05904 (accessed on 14 April 2025)).

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
