# Peer review of "A Clinical Validation of a Diagnostic Test for Esophageal Adenocarcinoma Based on a Novel Serum Glycoprotein Biomarker Panel: PromarkerEso"

_proteomes, 2025, doi:10.3390/proteomes13020023_

Round 1

Reviewer 1 Report

Comments and Suggestions for Authors

Attached an an attachment

Reviewer 2 Report

Comments and Suggestions for Authors

This manuscript presents the validation of a diagnostic test for esophageal adenocarcinoma based on the measurement of several serum glycoproteins. This pathology is one of the most prevalent malignancies worldwide, with limited diagnostic alternatives. The newly developed test was assessed on three esophageal adenocarcinoma cohorts - one for training, two for validation.

The introduction is well-structured and provides a coherent background that effectively contextualizes the study. Materials and methods are described adequately, with enough details to be able to reproduce the experiments. Results are clearly presented, and the Discussion is adequate. The authors have outlined the limitations of this study in a separate paragraph. Appropriate statistic models have been used to address the significantly different demographic distributions. 

I my opinion this study can be accepted for publication after making the following modifications/clarifications:

1. Introduction, lines 32-33: the authors should update esophageal cancer statistics based on the latest 2022 GLOBOCAN study (https://doi.org/10.3322/caac.21834)

2. Lines 67-70: To strengthen further the introduction, the authors are encouraged to briefly elaborate on the advantages of the methodology proposed- lectin-based magnetic bead array (LeMBA) pulldown method coupled with targeted mass spectrometry. Additionally, a brief mention of the technique's limitations would provide a more balanced perspective.

3. Lines 118-120: Authors should clarify whether the magnetic beads were removed following digestion quenching by formic acid and prior to LC-MS injection, and if this acidic solution was directly injected into the LC. 

4. Line 336: Blood drawn by venipuncture is not usually considered minimally invasive (for example a finger prick could be considered minimally invasive). I suggest replacing "minimally" with "less".

Reviewer 3 Report

Comments and Suggestions for Authors

In this manuscript, the authors introduced a novel glycoprotein biomarker panel test for esophageal adenocarcinoma (EAC), named PromarkerEso. The authors further validated this panel by measuring the relative concentrations of these serum glycoproteins via targeted mass spectrometry, and categorize cohort samples by the risk of EAC. The high sensitivity, specificity, and success rate of this panel test highlight its potential as an alternative non-invasive diagnostic tool for EAC.

The reviewer has a couple of comments to help improve the readability and rigor of this manuscript.

  1. One of the major comments is that, the authors need to emphasize the novelty of this work in the Abstract, at the end of the Introduction, and Discussion. Since this panel of biomarkers has been identified previously by Shah et al., it is necessary to differentiate what novel work is accomplished in this manuscript.
  • Jacalin was selected for this assay development as representative of lectin pulldown of glycoproteins, how does this improve selectivity/specificity of EAC diagnosis?
  • One extra cohort (VCB) is included besides the PROBE-NET and Ochsner cohorts which have been assayed by Shah et al. What extra aspects could this cohort add to this study? Why is the general population control (GenPop) important and relevant?
  • Are there any optimization on the workflow of targeted mass spectrometry?
  • Any considerations on statistical analysis that are different than what was used before?

  1. Please briefly introduce the definition of GenPop and its role as controls in this work, as compared to negative controls (NC).

In Figure 4, both NC and GenPop are represented as black dots, does this mean that they serve as similar roles in comparison with EAC samples?

  1. It is suggested that the authors clarify the definition of PromarkerEso, and different glycoproteins used in different stages (development, validation, etc.) of it.

So far three different combinations of glycoproteins are demonstrated:

  • 4 glycoproteins: alpha-1-antitrypsin, alpha-1-antichymotrypsin, complement C9 and plasma kallikrein. This set is supposed to be the PromarkerEso panel.
  • 6 glycoproteins: alpha-1-antitrypsin, complement C9, haptoglobin, hemopexin, PON1, KLKB1. This set is shown in Figure 2.
  • 9 glycoproteins, shown in line 176, line 208, and SI Table S5.

Please clarify the differences of these combinations of glycoproteins.

Some minor revisions:

  1. Line 72, line 85, and line 87.

Explaining “negative controls (NC)” one time is sufficient, and it should be placed the first time this term is used, which is line 72. “NC” could be directly used in line 85 and 87.

  1. Similarly as above, “general population (GenPop)” should be placed in Line 72.

  1. Add more keywords, such as Esophageal adenocarcinoma (EAC), Gastrointestinal diseases, targeted mass spectrometry, cancer diagnosis, PromarkerEso, etc.

  1. Please include specific clinical factors (age, sex and BMI) in the abstract and at the end of the introduction.

Round 2

Reviewer 1 Report

Comments and Suggestions for Authors

No comments